# Complex *k*-uniform tilings by a simple bitopic precursor self-assembled on Ag(001) surface

Lukáš Kormoš[1], Pavel Procházka [1], Anton O. Makoveev[1] & Jan Čechal [1,2✉]

The realization of complex long-range ordered structures in a Euclidean plane presents a significant challenge en route to the utilization of their unique physical and chemical properties. Recent progress in on-surface supramolecular chemistry has enabled the engineering of regular and semi-regular tilings, expressing translation symmetric, quasicrystalline, and fractal geometries. However, the *k*-uniform tilings possessing several distinct vertices remain largely unexplored. Here, we show that these complex geometries can be prepared from a simple bitopic molecular precursor – 4,4'-biphenyl dicarboxylic acid (BDA) – by its controlled chemical transformation on the Ag(001) surface. The realization of 2- and 3-uniform tilings is enabled by partially carboxylated BDA mediating the seamless connection of two distinct binding motifs in a single long-range ordered molecular phase. These results define the basic self-assembly criteria, opening way to the utilization of complex supramolecular tilings.

---

[1] CEITEC – Central European Institute of Technology, Brno University of Technology, Purkyňova 123, 612 00 Brno, Czech Republic. [2] Institute of Physical Engineering, Brno University of Technology, Technická 2896/2, 616 69 Brno, Czech Republic. ✉email: cechal@fme.vutbr.cz

Tessellation of a Euclidean plane into regular polygons has fascinated people since ancient times, and is highly relevant to mathematics, aesthetics, and crystallography[1]. In his rigorous description in 1619, Johannes Kepler introduced 11 edge-to-edge tessellations of a Euclidean plane by regular polygons, where all vertices of the tiling—i.e., joints of regular polygons—are of a single type[2]. These are called semiregular or Archimedean tilings (AT). Three of them—regular tilings—consist of only one kind of polygon (i.e., triangle, square, or hexagon) and the remaining 8 comprise a combination of two or more different polygons. Relaxing the condition of a single vertex type (or, more precisely, vertex isogonality, i.e., vertex equivalency by a symmetry of the tiling) generalizes the tessellation to $k$-uniform tilings, which may contain $k$ distinct types of vertices, or, more precisely, be $k$-isogonal[1].

Tilings by regular polygons are of particular interest because they appear to be an extremal solution to various problems and the treatment of more general problems can be often reduced to ones involving regular polygons[1]. The introduction of a few new tiles into tiling may profoundly alter its nature, which may be regarded as an analogue of the physical effects of introducing foreign atoms into crystal, and many crystallographic ideas (e.g., 'fault lines', 'planes of cleavage') seem to have their analogues in the theory of tilings[1]. Tilings thus present intriguing model systems for complex physical problems. In this respect, the supramolecular rhombus tilings, which provided insights into the physics of dynamically arrested systems and the role of entropy in the balance between order and randomness in molecular phases, serve as an illustrative example [3–6].

The recent progress in the experimental realization of complex surface tessellations on an atomic and molecular level[7–14] is driven by the intriguing physical[15–22] and chemical[23] properties of these systems. In this respect, the supramolecular chemistry offers tools for engineering of distinct self-assembled surface geometries that present an expression of semiregular[24–30], fractal[31–35], quasicrystalline[36–39], and random[3–6] tilings. Despite this effort, complex $k$-uniform tilings comprising a higher number of vertex types remain largely unexplored.

Here we describe the experimental realization of a long-range ordered system exhibiting 2-uniform and 3-uniform tilings (see Fig. 1a–c) self-assembled from a simple bitopic molecular precursor: 4,4'-biphenyl dicarboxylic acid (BDA, Fig. 1d). The higher degree of complexity is reached by its directed chemical transformation (carboxylation), enabling the realization of a series of stoichiometric mixtures of pristine, partially or fully carboxylated BDA, each exhibiting unique $k$-uniform tiling.

## Results

**Synthesis of tilings.** To synthetize complex tilings, BDA molecules were assembled on a four-fold symmetric Ag(001) substrate. The BDA molecule features two carboxylic end-groups that mediate intermolecular hydrogen binding and enable formation of extended supramolecular assemblies. These groups can be chemically transformed on metal substrates: sample annealing to elevated temperatures leads to carboxylation (also called deprotonation)—dissociation of hydrogen from carboxylic groups of BDA, which can be controlled by annealing temperature and time (Fig. 1e). To achieve the full control of the transformation the sample is monitored by low-energy electron microscopy (LEEM) and low-energy electron diffraction (LEED) during the annealing. In this way a homogeneous coverage of the sample with a single well-defined BDA phase is obtained. Molecular phases presented here are formed by annealing at temperatures in the range of 400 K–430 K. In the following, we will present the phases in order of increasing complexity bearing in mind that they are experimentally realized in a reverse order during the annealing.

Employing scanning tunneling microscopy (STM) and X-ray photoelectron spectroscopy (XPS) we have identified several long-range ordered periodic arrangements (molecular phases) depending on the ratio of BDA with a distinct level of carboxylation (Fig. 2). In the following, we denote the pristine BDA molecule 2H-BDA, as it still possesses 2 hydrogen atoms in each of the two carboxylic groups (2H atoms in total); correspondingly, 1H- and 0H- denotes BDA with only one or zero carboxylic hydrogens, respectively. The structure of the resulting molecular phases traced by STM is depicted in Fig. 3. Careful inspection of their geometry reveals that each of these structures represents a distinct tiling of the Euclidean plane introduced in Fig. 1a–c. The tiling can be described by the set of integers $\left[n_1^{\alpha_1} . n_2^{\alpha_2} . \ldots . n_n^{\alpha_n}\right]$ that correspond to the numbers of sides

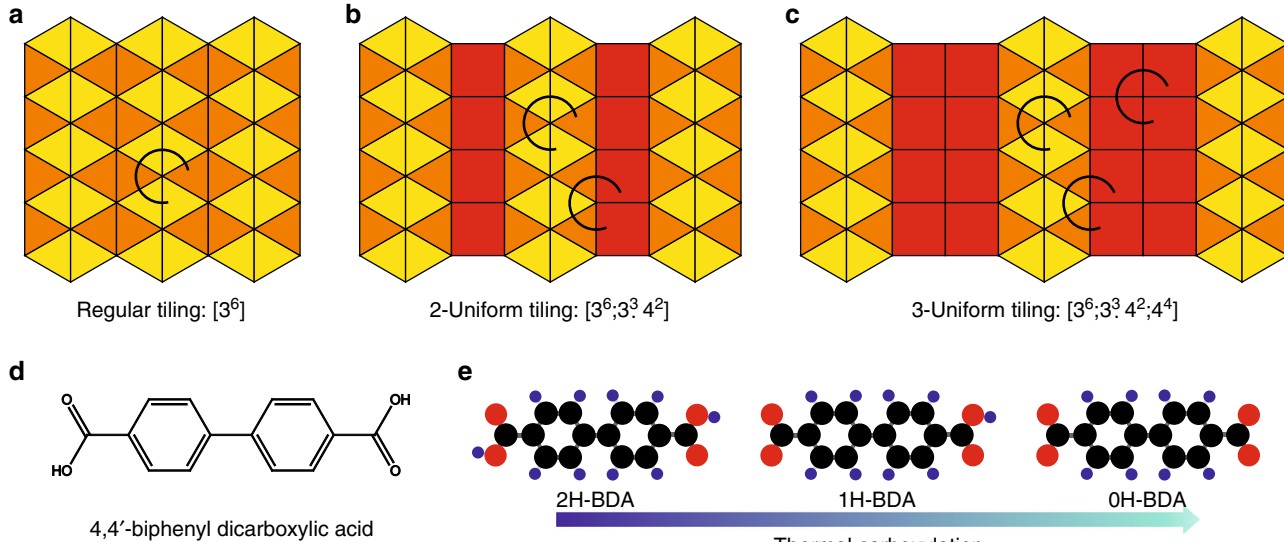

Regular tiling: [$3^6$]

2-Uniform tiling: [$3^6$; $3^3 4^2$]

3-Uniform tiling: [$3^6$; $3^3 4^2$; $4^4$]

4,4'-biphenyl dicarboxylic acid

2H-BDA          1H-BDA          0H-BDA

Thermal carboxylation

**Fig. 1 Uniform tilings and employed molecular precursors. a–c** Example of regular and $k$-uniform tilings of Euclidean plane comprising two type of tiles: triangles and squares. The distinct types of vertices are marked by arcs. There is one type of vertex in regular tiling (**a**), two types in 2-uniform (**b**), and three types in 3-uniform tiling (**c**), respectively. **d** Chemical structure of 4,4' biphenyl dicarboxylic acid (BDA) and (**e**) models of BDA with distinct level of carboxylation.

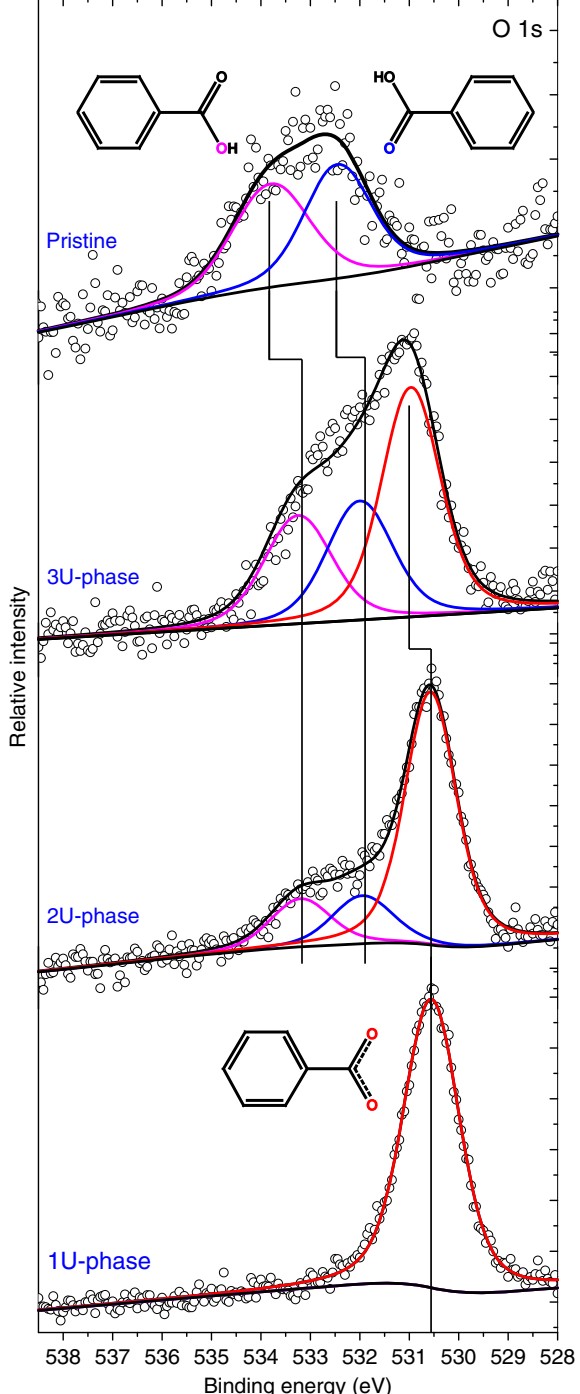

**Fig. 2 XPS analysis.** Detailed O 1s spectra measured on samples with BDA with 1U- to 3U-phases and a molecular phase comprising pristine BDA molecules. Spectra were acquired in high magnification mode using the pass energy of 20 eV integrating up to 180 sweeps with 0.1 s dwell time and 0.05 eV energy step. See Supplementary Discussion, X-ray photoelectron spectroscopy analysis, for details on XPS analysis.

$n_i$ of the polygons that meet at each vertex; the sequence of polygons is listed in a clockwise direction[1]. If two or more identical polygons are neighboring their count is given by exponent $\alpha_i$.

**1U-phase.** The simplest tiling is expressed by 0H-BDA molecules (1U-phase, Fig. 3a): here, six triangles meet at each vertex,

therefore the notation reads [3⁶]. The triangular tiles express the binding motif of three 0H-BDA molecules where molecule centers present the vertices of the tiling. Here the carboxylate groups point towards the benzene ring of neighboring BDA as depicted in Fig. 4a; this binding motif is typical for carboxylates on Ag surfaces[40]. We can recognize two kinds of triangular tiles portrayed by distinct color in Fig. 3a. The molecular structures associated with these triangles display a distinct chirality of BDA arrangement as detailed in Fig. 3b,c. XPS reveals a single O 1s component associated with a fully carboxylated BDA molecules at 530.7 eV; in agreement with the relevant systems[41]. The single component is consistent with a symmetric binding environment of both carboxylate oxygen atoms. This phase forms micrometer-sized single domain islands as shown in Supplementary Discussion, Long-range order over large areas.

A detailed analysis of diffraction patterns (Supplementary Discussion, LEEM analysis of molecular phases) and drift corrected STM images (Fig. 3) enables the proposal of the model given in Fig. 4a. The respective unit cell is commensurate with the Ag(001) substrate and can be expressed in matrix notation as $\begin{pmatrix} 3 & 5 \\ -3 & 2 \end{pmatrix}$. The matching of the rotated unit cell with the substrate causes a small distortion of a triangular tile from an ideal equilateral triangle. The measured lengths of sides (10.4 Å; 10.1 Å; 9.7 Å) and inner angles (60.3°; 56.3°; 63.4°) are close to the equilateral triangle. Importantly, all triangles are of an identical geometry, which also includes the two chiral structures (portrayed in yellow and orange). As the difference of the tiles from the ideal shape is minor, the structures introduced here present good approximants of Archimedean and $k$-uniform tilings[11,13,25,31,42].

**2U-phase.** Adding 1H-BDA molecules, and thus changing the composition of the system to a 2:1 ratio of 1H- and 0H-BDA, results in a new, long-range ordered, stable phase (2U-phase) presented in Fig. 3d that homogeneously covers the sample surface (see Supplementary Discussion, Long-range order over large areas, for details). The unit cell of the 2U-phase reads $\begin{pmatrix} 5 & 7 \\ -3 & 2 \end{pmatrix}$ and is commensurate with the substrate. This molecular phase comprises a new binding motif: complementary hydrogen bonding of two pristine carboxyl groups of neighboring 1H-BDA[40]. Presence of the intact carboxyl groups was confirmed by XPS. The XPS spectra in Fig. 2 shows two additional components the origin of which can be explained by comparison with the reference sample comprising only the pristine BDA molecules. The O 1s peak measured on the reference phase can be deconvoluted into two components that are associated with hydroxyl (533.8 eV) and carbonyl (532.45 eV) oxygen atoms of carboxylic group[41,43]. The intensity of these components is close to 1:1, which is consistent with the presence of the intact carboxyl groups. For the 2U phase, we observe peaks associated with both carboxylate and hydroxyl- and carbonyl-associated O1s peak components. The ratio of the peak intensities can be used to determine the ratio of functional groups present in each phase. According to the proposed model, the 2U-phase should display 0:2:1 ratio of 2H-:1H-:0H-BDA; this gives 2 carboxylic and 4 carboxylate groups, and on the level of O atoms it is 2 hydroxyl, 2 carbonyl and 8 carboxylate oxygens, i.e., ratio (1+1):4. The measured ratio of relative intensity of (hydroxyl + carbonyl): carboxylate peak components in 2U-phase is (0.15+0.15):0.7; this gives (1+1):4 ratio of carboxyl:carboxylate moieties matching the proposed model.

The bifunctional 1H-BDA possessing both carboxylic and carboxylate end-groups mediate a homogeneous transition between

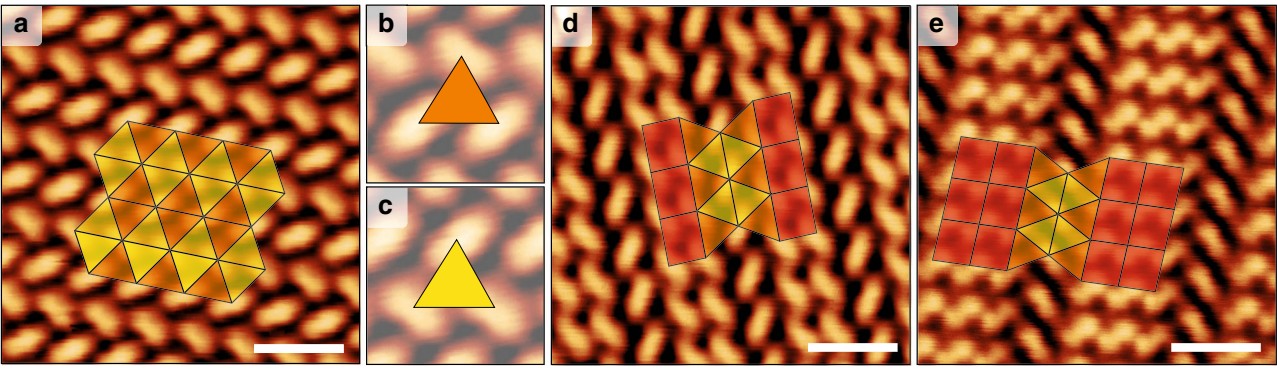

**Fig. 3 Real space STM view of self-assembled molecular networks.** These networks comprise BDA with distinct degree of carboxylation: pristine 2H-BDA, semi-carboxylated 1H-BDA and fully carboxylated 0H-BDA. **a** Regular tiling associated with a molecular phase comprising only 0H-BDA with tiling overlaid. **b**, **c** Detailed view of intermolecular binding motif associated with triangular tiles. **d** 2-uniform tiling realized by 2:1 mixture of 1H- and 0H-BDA. **e** 3-uniform tiling realized by 1:2:1 mixture of 2H-, 1H-, and 0H-BDA. Scale bar 2 nm.

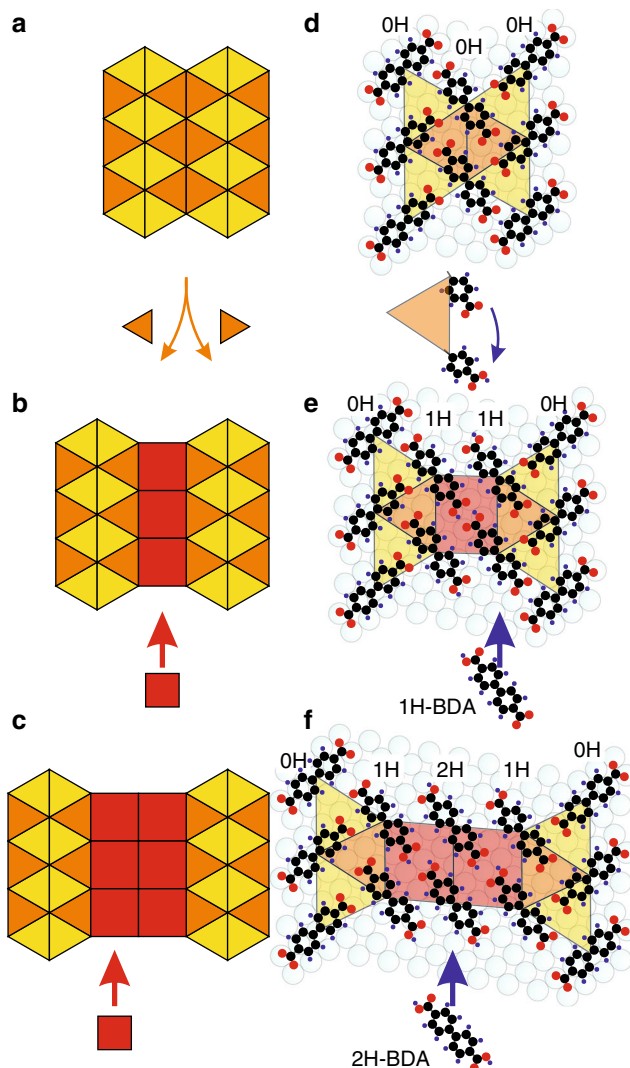

**Fig. 4 The increase of tessellation complexity by inserting new tiles and binding motifs.** **a–c** The 2-uniform tiling (**b**) is obtained by inserting a square tile between the rows of orange triangles of regular tiling (**a**). Likewise, insertion of an additional square results in the 3-uniform tiling (**c**). **d–f** Tentative molecular models based on LEED and STM data with colored tiles in the background. C: black, O: red, H: blue. Vertical rows of 0H-, 1H-, and 2H-BDA are marked by an inscription 0H, 1H, and 2H, respectively.

the new binding motif and the one in the 1U-phase: the 1H-BDA are thus seamlessly incorporated within the structure, adding the rectangular tiles as sketched in Fig. 4b. The position of 0H-BDA, strongly bound to the surface, is retained (see model in Fig. 4b). The molecular structure now comprises two distinct types of vertices (Fig. 1b), which is characteristic for the 2-uniform tiling [$3^6$; $3^4.4^2$]. Careful inspection reveals that the rectangular tile is in fact a parallelogram with sides 10.4 Å and 8.3 Å, and inner angles 88.2 ° and 91.8 °. As the unit cell is commensurate with the substrate, and the structure also adapts to a new binding mode, a slight change of triangular tile is also observed: sides and angles become (10.4 Å; 9.6 Å; 9.9 Å) and (56.5 °; 59.2 °; 64.3 °), respectively. Similarly to the 1U-phase, the distortion is caused by matching the 1U-phase unit cell to the substrate. The slight difference in the geometry between 1U and 2U phases points to the structural adaptability within employed molecular system, which is important in facilitating the formation of the molecular phase comprising two distinct binding motifs.

**3U-phase.** There is one additional phase with a chemical composition of a 1:2:1 ratio of 2H-:1H-:0H-BDA that exhibits the long-range order (denoted as 3U-phase) shown in Fig. 3e; the unit cell of the 3U-phase in matrix notation is $\begin{pmatrix} 8 & 10 \\ -3 & 2 \end{pmatrix}$. In this phase, the new component—2H-BDA—is incorporated to the structure in between the carboxylic end-group of 1H-BDA via complementary hydrogen binding motif as depicted in Fig. 4c. One unit cell of this phase should possess 3 carboxylic and 3 carboxylate groups. The (hydroxyl + carbonyl):carboxylate ratio measured by XPS is (0.26+0.27):0.47, which is close to theoretical ratio of 1:1 expected for this phase. The addition of 2H-BDA component results in an additional row of rectangles in the structure (see Fig. 4c) and also new vertex type: $4^4$. As the resulting structure comprises 3 distinct types of vertices, it presents a good approximant of a 3-uniform tiling noted [$3^6$; $3^4.4^2$; $4^4$]. Compared to 2U-phase the sizes of rectangular tiles became more uniform (10.4 Å and 9.2 Å) and slightly distorted (inner angles 85.0 ° and 95.0 °). The detailed geometry of the triangular tile also adjusts (10.4 Å; 10.2 Å; 11.0 Å, 56.9 °; 64.4 °; 58.7 °) to allow the matching the molecular structures with the Ag(001) substrate. Hence, we expect that tuning the substrate and employed molecular precursor may lead to realization of k-uniform tilings of an ideal geometry.

The exact structure of the molecular phases critically depends on the composition of the system, i.e., the ratio of 2H-BDA, 1H-BDA, and 0H-BDA. If the chemical composition is equal to the ratio given above the long-range ordered molecular phases are

homogeneous: mesoscale LEEM analysis reveals single phase compact islands in 4 symmetry equivalent orientations (see Supplementary Discussion, LEEM analysis of molecular phases, for details). This is well fulfilled for 1U and 2U phases which are stable in relatively large temperature window. On the other hand, the most complex 3U-phase readily undergoes transition to 2U-phase. Accordingly, we observe alternating stripes of these two binding motifs (see Supplementary Discussion, Long-range order over large areas and Sequence of 2U- and 3U-phases, for details). The conversion of the 3U phase into 2U phase takes place within the existing islands without significant alteration of their position and shape (see Supplementary Discussion, Phase transformations, for details). During the phase transformation the domains of newly formed 2U phase replace the existing 3U phase. The transformation process is mediated by defects within "nanocavities" propagating within the BDA islands (see Supplementary Movie 1). During the transformation an alternating sequence of 3U and 2U phases is observed. As both 2U-and 3U- phases are commensurate with the substrate, and the adjacent unit cell sides are exactly matching, we expect a random (non-Markovian) sequence of these two phases.

**Origin of the tilings**. Finally, we discuss the origin of these complex tilings. Previously, the Archimedean (1-uniform) tilings were realized from a single molecule featuring either two distinct functional centers that formed a single binding motif in complex tesselation[25] or by mixing two molecular phases that resulted in forming a new vertex type not present in the pure phases[26]. Another approach is to employ a rare earth directed metal-organic assembly, where the central ion enables the adaptability of opening angles between two neighboring ligands[24] or distinct number of coordinated ligands at each of the central ions[36]. In our case, expression of complex tessellation is enabled by interweaving of two phases seamlessly connected by the bifunctional 1H-BDA. Molecular phases comprising only 2H- and 0H-BDA show distinct intermolecular binding motives characteristic for these pure phases. As 1H-BDA is structurally similar to both 2H- and 0H-BDA and possesses both carboxylate and carboxylic group it can be incorporated at the perimeter of both limiting phases. Consequently, a single phase comprising both binding motifs arranged in a strip-like fashion is formed. This observation is distinct from majority multicomponent systems[44] and systems featuring carboxylic acids and carboxylates in paticular[45,46] as there the components tend to segregate into isostructural phases forming separate islands/domains. An interesting phase comprising 1:2 ratio of 2H- and 1H-BDA coexisting with other phases was observed on Cu(111) substrate[47]. That phase presents a 2-uniform tiling [$4^4$; $3^3.4^2$]. Compared with Ag(001) substrate the $3^6$ vertex is missing as the 0H-BDA molecules form the separate islands. The substrate geometry therefore has the important role in obtaining higher order uniform tilings.

The bifunctional 1H-BDA presents a key component allowing the seamless connection of the two binding motifs observed in the complex structures. Our strategy of a partial on-surface chemical transformation of precursor molecule, therefore, presents a viable way of reaching complex geometries. Carboxylation presents a simple way to provide chemically distinct functional groups and, to a large extend, keeps the molecular geometry intact, enabling the incorporation of several binding motifs into a single phase. This is also stressed by a recent study where stepwise deprotonation of 4,4′-dihydroxybiphenyl leads to a rich variety of tessellations[30]. Importantly, in contrast with Archimedean tilings comprising metal-organic networks[24,25,36], the tilings featuring hydrogen bonds presented here display nearly perfect long-range mesoscale order on micrometer length-scale.

## Discussion

In this communication we have shown a route to assembling complex k-uniform tiling approximants from a simple molecular precursor comprising two carboxyl groups on a biphenyl backbone. The complexity is reached by chemical transformation—carboxylation—of BDA, which results in a mixture of chemically distinct but structurally very similar molecules. Importantly the BDA bearing one carboxyl and one carboxylate is capable of mediating the seamless incorporation of two distinct binding motifs in one molecular phase enabling expression of higher order k-uniform tilings. This discovery defines design rules to reach complex structures and presents an important step in harnessing their intriguing physical and chemical properties.

## Methods

**Experimental procedures**. LEEM/STM/XPS experiments were carried out in a complex ultrahigh vacuum (UHV) system installed at the CEITEC Nano Research Infrastructure. Samples were prepared and analyzed in separate chambers between which samples can be transferred through a transfer line under UHV conditions (base pressure $2 \times 10^{-10}$ mbar). During the transfer (60–150 s), the pressure is slowly increased up to $2 \times 10^{-9}$ mbar and quickly restored to base level when the movement had ceased.

**Sample preparation**. Ag(001) single crystals (Mateck) were cleaned by repeated cycles of Ar$^+$ sputtering and annealing at 550 °C followed by a slow cooling to the room temperature in the Preparation Chamber with a base pressure of $2 \times 10^{-10}$ mbar. BDA molecules were deposited in the Deposition Chamber by the near ambient temperature effusion cell (Createc) from an oil heated crucible held at 185 °C on the sample held at room temperature. Submonolayer coverages (~40% of the surface) were obtained by BDA deposition for 5 min at a pressure lower than $8 \times 10^{-10}$ mbar. BDA was purchased from Sigma−Aldrich (97% purity) and used after thorough degassing in UHV.

**Scanning tunneling microscopy**. Scanning Tunneling Microscopy (STM) images were recorded with a commercial system Aarhus 150 (SPECS) equipped with Kolibri Sensor featuring a tungsten tip. Images presented in this work were measured at room temperature in the constant current mode; sample bias voltage was set between −150 mV and −300 mV and tunneling current to 60–100 pA. Distortion in the STM images was corrected assuming linear thermal drift of the sample derived from a series of consecutive images.

**Low-energy electron microscopy/diffraction**. Low Energy Electron Microscopy/ Diffraction (LEEM/LEED) experiments were carried out in a SPECS FE-LEEM P90 instrument with a base pressure of $2 \times 10^{-10}$ mbar. A bright field image was formed by the detection of electrons with an energy of 3 eV from the (0,0) diffracted beam. The diffraction pattern was collected from the area of $15 \times 10\ \mu m^2$ and microdiffraction pattern from the area befined by a 185 nm e-beam spot size on the sample.

**X-ray photoelectron spectroscopy**. X-ray Photoelectron Spectroscopy (XPS) analysis was performed on SPECS system equipped with Phoibos 150 spectrometer. Non-monochromatized Mg Kα radiation and normal emission geometry (emission angle 0°) was employed for all the measurements.

## Data availability

The primary datasets generated during the current study are available in the Zenodo repository, dx.doi.org/10.5281/zenodo.3690050 [48].

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

## Acknowledgements

This research has been financially supported by the Ministry of Education, Youth and Sports of the Czech Republic under the project CEITEC 2020 (LQ1601) under the National Sustainability Programme II and project CEITEC Nano+ (CZ.02.1.01/0.0/0.0/ 16_013/0001728) under programme OPVVV. This research was in part supported by the Horizon 2020 (project SINNCE, No. 810626) and GAČR (project No. 19-01536S). The research was carried out using the CEITEC Nano Research Infrastructure (MEYS, 2016–2019).

## Author contributions

L.K. prepared the samples, performed STM analysis and evaluated the data. P.P. prepared the samples, did LEEM/LEED experiments, evaluated the data and designed the models. A.M. prepared the samples and was involved in analysis. J.Č. designed the experiment, participated on experiment and data evaluation, interpreted the results, and wrote the manuscript.

## Competing interests

The authors declare no competing interests.
