## [Peer Review File · Nature Communications]

Reviewers' comments:

Reviewer #1 (Remarks to the Author):

Čechal et al used STM to study molecular assembly of 4,4'-biphenyl dicarboxylic acid (BDA) on a Ag(100) surface. Depending on the chemical states, i.e., step-wise deprotonation of the carboxylic groups, BDA molecules form different structures. The authors ascribed these structures as k-uniform tilings.

In reviewer's opinion, however, this claim is over-stated. In standard language of the 2D molecular assembly community, what they report is simply close-packed molecular organizations stabilized by inter-molecular hydrogen bonds, which is a phenomenon well-studied for 20 years. The step-wise deprotonation induced structure transformation has also been reported previously. The reviewer does not see a genuine relationship between the data and the interpretation which is in the framework of tiling. Thus the significance of this work is in doubt. In this regard, the reviewer cannot support its publication in NC.

Reviewer #2 (Remarks to the Author):

The authors report the on-surface design of complex k-uniform tilings by exploiting carboxylic ditopic species on Ag(001) and decarboxylation through thermal annealing.

They observe the formation of three different k-tilings going from regular to 2-uniform to finally 3-uniform tiling.

I like the manuscript, it is well written and the results are sound.

However, the authors claim the presence of three distinct carboxylic species (2H-BDA, 1H-BDA or OH-BDA), depending on conditions of annealing, without providing evidence by XPS or STS. Thus I will not recommend publication till such an evidence is provided.

Reviewer #3 (Remarks to the Author):

The paper proposes that the supramolecular assemblies formed by 4,4'-biphenyl dicarboxylic acid (BDA) and the carboxylated BDA on Ag(001) enable expression of complex k-uniform tilings. Namely, 2-uniform tiling was realized by 1:1 mixture of partially carboxylated BDA (1H-BDA) and fully carboxylated BDA (OH-BDA), and 3-uniform tiling was realized by 1:2:1 mixture of pristine, 1H-, and OH-BDA molecules. The observed long-range ordered molecular phase were attributed to the controlled carboxylation reaction and the distinct intermolecular binding between BDA. Overall the manuscript is well written with a persuasive model. I would like to recommend its publication if the authors could further clarify the following issues.

1. I believe the observed complex tilings are resulted from the combination of BDA and underlying Ag(001) substrate. BDA has been widely studied molecule and so is carboxylic acid group, what is their unique role in the presented tiling? What is the general applicability of the self-assembly criteria illustrated by BDA?
2. The XPS spectra should be more carefully explained. Are there atoms incorporated with carboxylated BDA in the self-assembly?
3. I would like to ask for more details about LEED pattern and its simulation. How is the adsorption site determined in the calculation?
4. Apparently, different rows of BDA undergo different carboxylate reaction in 2U- and 3U- phases, i.e. partial versus full carboxylation. This is intriguing and not explained.
5. The 2U- and 3U- phases are both commensurate with the substrate, therefore a random sequence of these two phases in the assembly is expected. It is not possible to obtain a pure phase using BDA or its carboxylated derivatives. The implication of molecular self-assembly in complex k-uniform tilings is arguable.

Reviewer #4 (Remarks to the Author):

The authors report an investigation of fabricating complex k-uniform tilings by utilizing a simple bitopic molecular precursor (4,4'-biphenyl dicarboxylic acid, BDA) on Ag(001) surface. Through the controlled chemical transformation of the precursor, long-range ordered structures exhibiting 2-uniform and 3-uniform tilings have been achieved. The structures of those tilings are well studied with STM, LEEM and XPS. The formation routines of those tilings are proposed based on above experiments. The manuscript is nicely written and the conclusions supported by the experiments and analyses.

However, the deprotonation of the carboxylic acid groups, which results in different self-assembly structures, has been used and studied in the on-surface construction of supramolecular structures. Moreover, there is a similar supramolecular self-assembly using the same precursor has been reported previously (ref 44). In comparison with previous and recent reported molecular tilings on surfaces (ref 25, 26), the analysis of the k-uniform tilings appears in a preliminary status, and the novelty of this work do not merit consideration in the high standards of Nature Communications.

So, I cannot recommend the publication of the present submission in Nature Communications.

Reviewers' comments:

Reviewer #1 (Remarks to the Author):

Čechal et al used STM to study molecular assembly of 4,4'-biphenyl dicarboxylic acid (BDA) on a Ag(100) surface. Depending on the chemical states, i.e., step-wise deprotonation of the carboxylic groups, BDA molecules form different structures. The authors ascribed these structures as k-uniform tilings.

In reviewer's opinion, however, this claim is over-stated. In standard language of the 2D molecular assembly community, what they report is simply close-packed molecular organizations stabilized by inter-molecular hydrogen bonds, which is a phenomenon **well-studied for 20 years**. The step-wise deprotonation induced structure transformation has also been reported previously. The reviewer does not see a genuine relationship between the data and the interpretation which is in the framework of tiling. Thus the significance of this work is in doubt. In this regard, the reviewer cannot support its publication in NC.

Response:

The reviewer is correct in statement that the self-assembly is studied for 20 years and some work was also devoted to deprotonation. However, in our opinion, the current level of understanding of self-assembled systems is far from complete with significant aspects and implications overlooked. Tessellation of Euclidean plane present a fresh view on the structure of molecular systems. Unique properties associated with complex geometrical structure may point to long-sought applications of self-assembled systems. After submission of our work, a paper in which tilings represented by similar molecular precursor (4,4'-dihydroxybiphenyl) formed by sequential deprotonation was published in ACS Nano:

L. Feng, T. Wang, Z. Tao, J. Huang, G. Li, Q. Xu, S. L. Tait, J. Zhu: Supramolecular Tessellations at Surfaces by Vertex Design, *ACS Nano* **2019**, 13, 10603–10611.

Hence, contrary to reviewer's view, we think that employing the simple systems may still provide significant new knowledge and reveal fundamental properties of the self-assembled systems and identification of particular geometries may open a way for harnessing properties associated with the non-trivial structure.

Reviewer #2 (Remarks to the Author):

The authors report the on-surface design of complex k-uniform tilings by exploiting carboxylic ditopic species on Ag(001) and decarboxylation through thermal annealing.

They observe the formation of three different k-tilings going from regular to 2-uniform to finally 3-uniform tiling.

I like the manuscript, it is well written and the results are sound.

However, the authors claim the presence of three distinct carboxylic species (2H-BDA, 1H-BDA or OH-BDA), depending on conditions of annealing, **without providing evidence by XPS or STS**. Thus I will not recommend publication till such an evidence is provided.

Response:

We have provided a thorough XPS analysis in the Supplementary Information. Whereas we think that its presence in the main text is not vital for the argumentation and text-flow, we have moved it into the main text for completeness of presentation.

Action: We have implemented the XPS in the main text.

Reviewer #3 (Remarks to the Author):

The paper proposes that the supramolecular assemblies formed by 4,4'-biphenyl dicarboxylic acid (BDA) and the carboxylated BDA on Ag(001) enable expression of complex k-uniform tilings. Namely, 2-uniform tiling was realized by 1:1 mixture of partially carboxylated BDA (1H-BDA) and fully carboxylated BDA (0H-BDA), and 3-uniform tiling was realized by 1:2:1 mixture of pristine, 1H-, and 0H-BDA molecules. The observed long-range ordered molecular phase were attributed to the controlled carboxylation reaction and the distinct intermolecular binding between BDA. Overall the manuscript is **well written with a persuasive model**. I would like to recommend its publication if the authors could further clarify the following issues.

1. I believe the observed complex tilings are resulted from the combination of BDA and underlying Ag(001) substrate. BDA has been widely studied molecule and so is carboxylic acid group, what is their unique role in the presented tiling? What is the general applicability of the self-assembly criteria illustrated by BDA?

Response:

We have a unique possibility to monitor the phase transition with LEEM, therefore we have identified the new phases and are able to prove that they are the only ones that are present at surface at given conditions.

In this way we were able to identify mixed phases comprising the semi-deprotonated (1H-BDA) molecules, which are capable to engage two distinct binding motives characteristic for pure and fully deprotonated phases. As a result, we observe a fascinating phase comprising stripe patterns of BDA molecules with distinct level of carboxylation with alternating bonding motives.

Action: We have provided a more detailed discussion on significance of bifunctional 1H-BDA.

2. The XPS spectra should be more careful explained. Are there adatoms incorporated with carboxylated BDA in the self-assembly?

Response:

As all the employed instruments are part of a complex UHV system, it allows the analysis of the same samples by distinct instruments. In addition to presented phases we have identified additional phases; some of them incorporating also Ag atoms. In this case we observe a different geometrical arrangement on molecules with Ag atoms clearly identified by STM. As the Ag containing phases show a distinct diffraction patterns they are easily recognized in diffraction. Before the XPS analysis, we carefully checked the sample by LEEM to ensure the phase uniformity at mesoscale level and exclude the presence of minor phases which would influence the XPS conclusions.

Action: We have provided a thorough description of XPS measurements including statement on possible presence of adatoms.

3. I would like to ask for more details about LEED pattern and its simulation. How is the adsorption site determined in the calculation?

Action: We have provided a detailed description of LEED pattern simulation and added a more detailed discussion of the 'local similarity approach' for determination of unit cell from diffraction spot rich multidomain diffraction patterns.

4. Apparently, different rows of BDA undergo different carboxylate reaction in 2U- and 3U- phases, i.e. partial versus full carboxylation. This is intriguing and not explained.

Response:

Also, in view of comments of reviewer #4 comments we have decided to add the LEEM real-time videos showing the transformation.

Action: Added a new section describing the phase transformation in real time.

5. The 2U- and 3U- phases are both commensurate with the substrate, therefore a random sequence of these two phases in the assembly is expected. It is not possible to obtain a pure phase using BDA or its carboxylated derivatives. The implication of molecular self-assembly in complex k-uniform tilings is arguable.

Response:

We are able to monitor the evolution of particular phases with LEEM at mesoscopic level. We did not observe formation of pure phases at this particular point of the phase transformation (when 3U, 2U phase were present). Of course, we observed pure phases after RT deposition of BDA (a well-known phase featuring complementary hydrogen bonding) and after the full carboxylation (1U phase).

Action: a more detailed discussion of pure phases was added to the manuscript.

Reviewer #4 (Remarks to the Author):

The authors report an investigation of fabricating complex k-uniform tilings by utilizing a simple bitopic molecular precursor (4,4'-biphenyl dicarboxylic acid, BDA) on Ag(001) surface. Through the controlled chemical transformation of the precursor, long-range ordered structures exhibiting 2-uniform and 3-uniform tilings have been achieved. The structures of those tilings are well studied with STM, LEEM and XPS. The formation routines of those tilings are proposed based on above experiments. **The manuscript is nicely written and the conclusions supported by the experiments and analyses.**

However, the deprotonation of the carboxylic acid groups, which results in different self-assembly structures, has been used and studied in the on-surface construction of supramolecular structures. Moreover, there is a similar supramolecular self-assembly using the same precursor has been reported previously (ref 44). In comparison with previous and recent reported molecular tilings on surfaces (ref 25, 26), the analysis of the k-uniform tilings appears in a preliminary status, and the novelty of this work does not merit consideration in the high standards of Nature Communications.

So, I cannot recommend the publication of the **present submission** in Nature Communications.

Response:

The previous study (ref. 44) indeed reported the phase similar to the one we identify as 3-uniform tiling on the Cu(111) surface where the mixture of several distinct phases was present. Actually, their phase, although visually similar, present a 2-uniform tiling $[4^4; 3^3.4^2]$ with 3^6 vertex missing; Importantly, in their paper the special geometry arrangements were not recognized. The comparison with our results obtained on Ag(001) with those on Cu(111) also highlights the importance of employed substrate and difficulties associated with synthesis of higher order uniform tilings.

To provide a deeper insight also into the fundamentals of self-assembly we have decided to include a LEEM real-time videos showing the transformation between all three phases considered here that we initially excluded as we intended to provide a clear view with a focus on the higher order uniform tiling. The videos show the internal transformation mechanism between 3U, 2U and 1U phases, where the transformation occurs in voids running through the macroscopically intact islands. We hope that in this way we provide a significant new insight also into the self-assembly besides the identification of new tiling geometries. We hope that the real-time view, a thorough discussion of XPS and unit cell determination from diffraction will rise the elaboration level of the manuscript.

Action: (1) Added a section on phase transformation and included supplementary video on the topic. (2) A thorough elaboration including XPS analysis a unit cell determination from multidomain LEED pattern was included into manuscript. (3) Discussion of phases on Cu(111) in the context of 2- and 3-uniform tilings.

Reviewers' comments:

Reviewer #2 (Remarks to the Author):

The authors have answered to the referees' comments in a proper way. Thus I will recommend publication as it is.

Reviewer #3 (Remarks to the Author):

The revised manuscript provides additional XPS and LEED results that give more insight into the self-assembly structures. The improvement is merited for acceptance.

Reviewer #4 (Remarks to the Author):

In the revised manuscript, the author studied the phases transformation of 3U to 2U to 1U through LEEM, which provides some new insights about the evolution of the tiling on surface. According to the definition of the k-uniform tiling by the authors based on the STM image, the similar k-uniform tiling has been found in the previous report (J. Am. Chem. Soc. 2013, 135, 7458-7461). Unfortunately, the internal transformation mechanisms between different phases cannot be well demonstrated by LEEM. Besides, the studies of well-known step-wise deprotonation (by XPS), which induced the evolution of hydrogen bonds, also can not improve the novelty of the manuscript. So, I still insist that the novelty of this manuscript does not meet the high standard of Nat. Commun. and I could not recommend its publication in this journal.

Reviewer #5 (Remarks to the Author):

I have looked through the paper and the previous exchanges between reviewers and authors. The paper is well written and provides a clear message about the observation of k-tilings. The novelty of the manuscript lies entirely with the classification of observed arrangements in terms of these tilings; as several referees have commented the the de-hydrogenation of carboxylic acids and the influence of this process on molecular ordering has been studied for many years and there is no intrinsic novelty of this aspect of the paper. So the question is whether the observation of this particular tessellation is enough to warrant publication in NC. I think I would support publication but the case is not overwhelming; in particular the recent ACS Nano Feng et al. mentioned by the authors is a very comprehensive piece of work which explores very similar ideas (although not the same particular arrangements).

However I had some other comments:

1. tessellations imply ordering over reasonably long length scales. The images shown in the main text are very small and could be due to localised domains rather than extended order. The authors should

include larger images, preferably in the main paper, but in the SI if that is preferred, to show the molecular ordering on a scale which is comparable to the current Fig S8.

2. The ACS Nano paper mentioned by the authors in their response should be cited.

3. I didn't understand the arguments related to the domain boundaries in Fig S8 - top of page 12 - dissolving of phases. Also there seemed to be regions of 4U (?) in Fig. S8 - do these extend over large areas.

4. In my view the LEEM images looked very interesting - I haven't seen this kind of data used to investigate the evolution of these phases previously. It seemed to be a shame not to highlight it more prominently.

Reviewers' comments:

Reviewer #2 (Remarks to the Author):

The authors have answered to the referees' comments in a proper way.
Thus I will recommend publication as it is.

Reviewer #3 (Remarks to the Author):

The revised manuscript provides additional XPS and LEED results that give more insight into the self-assembly structures. The improvement is merited for acceptance.

Reviewer #4 (Remarks to the Author):

In the revised manuscript, the author studied the phases transformation of 3U to 2U to 1U through LEEM, which provides some new insights about the evolution of the tiling on surface. According to the definition of the k-uniform tiling by the authors based on the STM image, the similar k-uniform tiling has been found in the previous report (J. Am. Chem. Soc. 2013, 135, 7458-7461). Unfortunately, the internal transformation mechanisms between different phases cannot be well demonstrated by LEEM. Besides, the studies of well-known step-wise deprotonation (by XPS), which induced the evolution of hydrogen bonds, also can not improve the novelty of the manuscript. So, I still insist that the novelty of this manuscript does not meet the high standard of Nat. Commun. and I could not recommend its publication in this journal.

Reviewer #5 (Remarks to the Author):

We thank the reviewer for careful reading of our manuscript and appreciation of our work. Further we are grateful for the remarks that helped us clarifying the message of the manuscript. Below we address his/her concerns.

I have looked through the paper and the previous exchanges between reviewers and authors. The paper is well written and provides a clear message about the observation of k-tilings. The novelty of the manuscript lies entirely with the classification of observed arrangements in terms of these tilings; as several referees have commented the the de-hydrogenation of carboxylic acids and the influence of this process on molecular ordering has been studied for many years and there is no intrinsic novelty of this aspect of the paper. So the question is whether the observation of this particular tessellation is enough to warrant publication in NC. I think I would support publication but the case is not overwhelming; in particular the recent ACS Nano Feng et al. mentioned by the authors is a very comprehensive piece of work which explores very similar ideas (although not the same particular arrangements).

However I had some other comments:

1. tessellations imply ordering over reasonably long length scales. The images shown in the main text

are very small and could be due to localised domains rather than extended order. The authors should include larger images, preferably in the main paper, but in the SI if that is preferred, to show the molecular ordering on a scale which is comparable to the current Fig S8.

Response:

The presented tessellations show long-range order and are observed in micrometer sized domains. We prepared the phases in LEEM to ensure the purity of the phases forming micrometer sized islands. Here the long-range order was confirmed by a sharp single-phase diffraction patterns taken from area ~ 150 square micrometers (Figures S2, S3, S4). The samples we subsequently transferred to STM for an analysis. Here we observed very large single-phase islands (new Figure S7) for 1U and 2U phases. As we mentioned in the manuscript (top of page 12) the 3U phase slowly transforms to a mixed phase at room temperature; hence in STM after a prolonged time we observed coexistence of the 3U phase with the mixed phase: new Figure S7c shows the boundary of 3U phase (bottom) with a mixed phase (top). So, in this case the minimal area of 3U phase observed in STM is $15 \times 50 \text{ nm}^2$.

Action: We have introduced a new section in Supporting Information (Section 3) and referenced it at relevant positions within the manuscript (pages 6, 8, and 12).

2. The ACS Nano paper mentioned by the authors in their response should be cited.

Response:

The paper is now properly cited.

Action: We have included the citation to introduction and provided following text on page 13: "This is also stressed by a parallel study where stepwise deprotonation of 4,4'-dihydroxybiphenyl leads to a rich variety of tessellations."

3. I didn't understand the arguments related to the domain boundaries in Fig S8 - top of page 12 - dissolving of phases. Also there seemed to be regions of 4U (?) in Fig. S8 - do these extend over large areas.

Response:

We agree with reviewer that the particular part on page 12 is not completely clear without providing a proper context of molecular phase transformation mechanism. We have rewritten this part together with the related text in Section 4 (former Section 3) of Supporting Information and caption of Figure S9 (former Figure S8); we hope that it is much clearer now.

Concerning the 4-Uniform tiling, we have indeed observed local inclusions displaying 4 2H-BDA molecules in a row. These can be associated with 4-Uniform tiling featuring 3 distinct vertices. As they present a minor inclusion observed only in the mixed phase near defect sites, we did not originally highlight their presence. We agree with the reviewer that its worth highlighting so we did it accordingly.

Action: We have rewritten the relevant text on page 12 of main article, in Section 4 (former Section 3) of Supporting Information, and Figure S9 (former Figure S8) caption.

4. In my view the LEEM images looked very interesting - I haven't seen this kind of data used to investigate the evolution of these phases previously. It seemed to be a shame not to highlight it more prominently.

Response:

We are also enthusiastic in the possibility to visualize phase transformations by LEEM. We are about to submit a study on the transformation of the intact molecular phase to an additional phase that is not included in this manuscript. To maintain the clear focus of the manuscript on the main topic, i.e. k -uniform tilings, we only included the data on phase transformation to Supporting Information as they affirm the interconnection between the 2U and 3U tilings.

Action: We kept the LEEM data on transformation in Supporting Information.